# Research on the Construction of Safety Information Ontology Knowledge Base and Accident Reasoning for Complex Hazardous Production Systems-Taking Methanol Production Process as an Example

**Meng Liu, Rui Huang * and Fangting Xu**

School of Resources and Safety Engineering, Central South University, Changsha 410083, China
* Correspondence: huangrui@csu.edu.cn; Tel.: +86-13787792851

**Abstract:** Taking methanol production as an example, the concept of "ontology" is introduced to construct a safety knowledge ontology, and a safety information knowledge base is created with the help of the Protégé software. These can be used to efficiently handle the massive safety information data of dangerous chemical enterprises, associate all kinds of miscellaneous information, and improve the level of safety management. An accident tree reasoning model is designed to determine the cause of the accident using accident tree reasoning, and to mine the vast knowledge of safety information, according to safety information knowledge and accident tree analysis theory. Using these methods, the storage, processing, and reuse of safety information are realized, the efficiency of safety management can be improved, and the defects caused by incomplete personnel knowledge structure can be avoided.

**Keywords:** accident tree; hazard enterprise; knowledge base; ontology; safety information; reasoning

## 1. Introduction

Big data platforms are increasingly being used by enterprises for production safety management in the context of information technology [1], and numerous academics have looked into this topic. For example, Zhang, H. [2], Qu, W.B. [3], and others have used IBM (Building Information Model) technology in the construction industry, which has improved the informatization level of construction safety management. Zhang, M.Y [4] proposed the use of sensor technology to achieve real-time construction safety management. The traditional manual input of safety information has gradually given way to an increasingly extensive and sophisticated set of safety data that is automatically collected by computers. In the course of production, businesses produce data in a variety of formats and from a wide range of sources. The growth of information technology has caused knowledge with potential value to be expressly or implicitly dispersed in large amounts of data. Enterprise safety management is hampered by the loss of a significant proportion of useful safety information due to imperfect information processing methods. First, the acquisition of safety information is incomplete. The volume of safety information data has dramatically expanded against the backdrop of safety production informatization. The lack of sufficient human and material resources prevents personnel from obtaining all safety information data in a timely fashion. Second, the way safety information is stored lacks systematization and is kept in a disorganized manner. To categorize and process the safety information, more regular and systematic storage methods are required, such as how to classify the safety information of the hazardous chemical industry. Third, the application of safety information is not sufficient. Managers frequently stop after analyzing the surface value of safety information and fail to further explore the deep value, such as how to uncover the knowledge on the causes of accidents. These defects in safety information management

have caused a large number of valuable safety information not to be applied, causing losses to safety management. Therefore, it is vital to create efficient methods for extracting crucial safety data and addressing the issue of safety management effectiveness. The key aim of this paper is to study the logical reasoning method and information correlation technology based on the knowledge base.

The knowledge base, as a repository of information, organizes knowledge, speeds up the flow of knowledge, and encourages knowledge sharing and exchange. Ontology is the explanation of objective facts. The term "ontology" was first used in the field of knowledge engineering in the late 1980s and early 1990s. It is possible to realize knowledge expression and inquiry by building an ontology-based knowledge model. Many sectors of the information processing industry decide to store, describe, and express information data using ontology-based knowledge bases: in the field of traditional Chinese medicine, Liao, CH, et al. [5] used BNF (Backus-Naur Form) to research and build a knowledge base system based on TCM ontology; in the field of transportation, Li et al. [6] built a case knowledge base for emergency handling of high-speed railway signaling equipment failures; Kou, Y.T et al. [7] applied ontology knowledge base theory to the agricultural field and created an ontology-based knowledge service platform in the agricultural field; and in the field of education, Sun, Y., etc. [8] proposed ontology-based domain knowledge representation to improve the sharing and reuse efficiency of multimedia textbooks.

The analysis of the aforementioned literature reveals that the knowledge base theory based on ontology has been utilized in many domains, widening the approach to information management and enhancing its effectiveness as well as information sharing and reuse. As a result, the ontological knowledge base offers special benefits for information management. Ontology theory has been used by many academics to build the knowledge base for information management. However, the application research of the ontology knowledge base is not sufficiently in-depth for safety information management in the field of hazardous chemicals. This paper is dedicated to the study of the precise collection and logical association technology of safety information in the field of hazardous chemicals. Taking methanol production as the research object, we propose building the safety information ontology knowledge base, and use the ontology language to realize the unified representation and storage management of safety information.

Due to their hazardous materials and energy intensity, dangerous chemical companies have a significant likelihood of encountering accident conditions [9,10]. Accidents involving hazardous chemicals are typically unexpected and catastrophic. For example, in 2005, the British Bunsfield oil storage depot exploded [11]; an ammonium nitrate explosion occurred in western Texas in April 2013 [12]; in August 2015, 165 people were killed in an explosion of a dangerous chemicals warehouse in Tianjin Port [13]; in March 2019, a hazardous chemical explosion occurred in Yancheng, Jiangsu, China, killing 78 people [14]. Due to the significant number of fatalities and property damages brought on by accidents involving hazardous chemicals, it is essential to assess the risks posed by this industry and establish strategies for preventing accidents. A probability index for accidents was proposed by Huh, D.A [15]. To assess the overall risk posed by chemicals and aid in the risk management of chemical plants, the accident probability index is calculated using four readily available alternative indicators, including chemical health hazard, chemical handling capacity, national chemical accident frequency, and plant accident frequency. In order to reduce the risk associated with the transportation of hazardous chemicals, Chang, L and colleagues constructed a site model of that risk, used the TruckSim software to simulate accidents involving hazardous chemicals, and performed a risk classification. The model is employed to assess the risk associated with the transportation of hazardous chemicals and to ensure the safety of the vehicles involved in such transportation [16]. Soltanzadeh, A investigated accidents in the chemical processing industry, used exploratory and confirmatory factor analysis (EFA and CFA) and structural equation modeling (SEM) for data analysis, and found that unsafe conditions and unsafe behaviors are the factors that directly affect the occurrence rate of chemical accidents [17]. It should be mentioned

that the majority of the current research on methods for chemical accident analysis and risk identification focuses on calculating parameter values and index selections to assess the risk of hazardous chemicals. However, it is unclear how to pinpoint accident causes with accuracy. In order to accurately implement preventative actions to avoid and control the incidence of accidents, it is required to develop new methods and technologies for analyzing and identifying the causes of accidents. This paper proposes combining ontology and reasoning, and design accident reasoning rules based on accident tree analysis theory, to mine accident causes.

## 2. Methods

### 2.1. Ontological Theory

Ontology is "a conceptual description of concepts in the domain and their relationships" [18]. Ontology knowledge representation is a kind of knowledge representation method that uses elements such as classes, attributes, relationships, functions, and axioms to abstract the essence of things [19]. The ontology-based knowledge representation method can effectively promote knowledge exchange and sharing, and has a strong reasoning ability.

At present, the method of using ontology to build a knowledge base is not yet fully mature. Most of the existing construction methods are summarized by scholars from specific ontology construction projects, which have distinctive domain characteristics. Common ontology knowledge base construction methods include the skeleton method, TOVE method, KACTUS engineering method, seven-step method, etc. The seven-step method is proposed based on the protégé ontology construction tool, which is relatively mature at present. Therefore, this paper focuses on the seven-step method to build the safety information ontology knowledge base.

The seven-step method is an ontology construction method proposed by Stanford University. Its specific construction process is as follows:

(1) Define the scope of domain knowledge. Define the research field, ontology category, users, and the functions to be implemented by the ontology.

(2) Reuse existing ontology. Check whether there are reusable ontologies. If so, refine and expand the existing ontology to reduce the cost of ontology development.

(3) Define key terms in the field. List the key terms in the research field to ensure the completeness of the terms.

(4) Define the classes and relationships in the ontology. Define the classes in the ontology based on the defined terms, and determine the relationships between the classes.

(5) Define the properties of the classes. Properties are the characteristics of the classes in ontology.

(6) Define the property facets. Define the properties in detail, including object properties and data properties.

(7) Create the instance. Add instances related to the class and their property relationships based on the created ontology.

### 2.2. SWRL (Semantic Web Rule Language) Reasoning Theory

SWRL is a language that presents rules in a semantic way. XML (Extensible Markup Language) and RDF (Resource Description Framework) are the two primary formats for expression. XML mode is described in RuleML + OWLX, and RDF mode is described in OWL + RDF [20]. These two approaches have different expressions but the same grammatical structure and function. The goal is to make ontology more useful by better integrating it with reasoning principles.

The basic form of an SWRL rule is to express the derivation relationship between the premise and the conclusion. Both the premise and the conclusion can include single or multiple basic propositions. The relationship between basic propositions is the relationship between logic AND [21]. The form is: Antecedent → Consequent. SWRL consists of Imp, Atom, Variable, and Building. Atom is used to define the restrictive formula of conditional

judgment; Imp is used to define rules. The restriction in Imp is provided by Atom; Building is used to define various logical comparison relationships, such as numerical comparison, Boolean operation, string operation, etc.

In ontology, SWRL rules [22] mainly use two restrictions:

① C (x): x represents the instance of variable or ontology; C is a class. This expression indicates that x is an instance under class C. For example, Mother *(? x)* represents an instance x under the class Mother.

② P (x, y): x, y represents the instance of variable or ontology, and P is the object property. For example, HasSister *(? x,?y)* indicates that y is the sister of x.

Typical SWRL rules are as follows:

$$Person(?x)^{\wedge}HasMother(?x,?y)^{\wedge}HasSister(?y,?z) \rightarrow HasAunt(?x,?z)$$

In this rule, x, y, and z are three different instances of the class Person, and "Has-Mother", "HasSister", and "HasAunt" are three different object properties. The premise of this rule is that if y is the mother of x and z is the sister of y. The conclusion is that z is the aunt of x.

### 2.3. Accident Tree Analysis Theory

Accident tree analysis theory is of great significance in safety science for analyzing the causes of accidents and formulating preventative measures. The accident tree analysis method is a logical analysis method of the causes and results of accidents using operational research principles. It mainly includes two elements: logic gates and events. Logic gates are used to represent the relationship between events. Events include basic events, middle events, and top events [23]. In the process of using the accident tree, the system to be analyzed must be defined first, and the accident to be analyzed (called the top event) must be found in the system as the starting point of analysis. Secondly, the cause of the top event is identified according to logical reasoning (called the middle event). Finally, logical reasoning continues to be used to find out the cause of the middle events until they can no longer be divided (called the basic event). The accident tree ontology has been expressed in the knowledge representation and storage parts. The accident tree ontology includes events (basic events, middle events, and top events), event states (happen and non-happen), logic gates (AND gate and OR gate), and corresponding property definitions for the accident tree ontology.

In accident tree analysis theory, logic gates connecting events mainly include AND gates and OR gates.

(1) AND gate: Only when all of the input events connected to the AND gate occur will the output event occur. The AND gate represents the intersection of events. As shown in Figure 1, logic gate $G_1$ is an AND gate, the connected events are input events A and B, and output event T. If event T occurs, events A and B must occur. That is, T = A∩B.

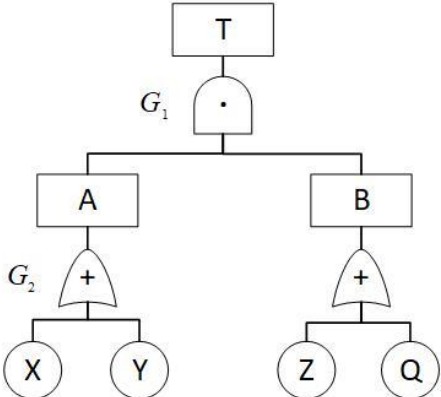

**Figure 1.** Accident tree to be analyzed.

(2) OR gate: Any input event that occurs in the input events linked to the OR gate will result in the output event also occurring. The OR gate represents the union of events. As shown in Figure 1, logic gate $G_2$ is an OR gate. The connected events are input events X and Y, and output event A. If event A occurs, the result may be: input event X occurs, Y does not occur; X does not occur, Y occurs; both X and Y occur. That is, A = X U Y. In this study, in order to improve the accuracy and conciseness of the reasoning model, the input event with the highest probability of occurrence is defaulted to occur, and the other events do not occur, so as to determine the most likely cause of the accident.

## 3. Overview of Methanol Production Safety Information Knowledge Base

To facilitate the research, this paper selects a methanol production process as the research object to analyze the safety information of methanol production. Methanol production is a fairly representative chemical production process in the field of hazardous chemicals. It possesses the typical traits associated with the production of hazardous substances, including flammability, explosiveness, toxicity, etc. The methanol production process can be dissected and examined since it includes safety information such as equipment, process flow, danger sources, accidents, etc. By looking for safety information management techniques for methanol production and building a safety information knowledge base, this paper aims to provide management examples and theoretical support for the safety information of the hazardous chemicals industry.

### 3.1. Methanol Production Process

The methanol production process in this study adopts a chemical synthesis method [24]. The process for producing methanol involves using coal as the raw material, creating raw gas using an integrated four-nozzle gasifier, adjusting the $H_2/CO$ ratios using a sulfur-resistant shift, removing sulfur and carbon using low-temperature methanol washing, recovering the sulfur using the Claus method, synthesizing the methanol using the low-pressure method, and distilling the crude methanol in three towers to produce refined methanol. Figure 2 depicts the precise process flow.

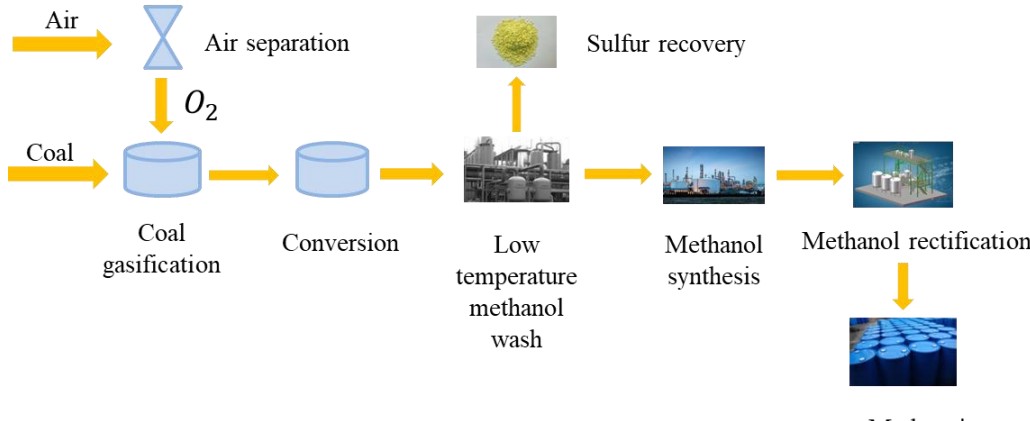

**Figure 2.** Methanol Production Process Flow Chart.

### 3.2. Node Analysis of Methanol Production Safety Information Generation

Safety information refers to the collection of information that plays a safety role in the labor production process [25]. Safety information's primary role is to support company safety management and prevent accidents. From the perspective of safety information application, combined with the methanol production process, the methanol production safety information is classified. Therefore, when building a knowledge base, safety information can be categorized into three groups:

(1) Production status information, such as whether or not tools and equipment are in good condition, how safe they are while in use, how the production process is flowing, and safety behavior information of operators, etc.

(2) Hazard source information involved in methanol production, including hazard sources existing in the human, material, environment, management, etc.

(3) Information on main accident types and prevention and control measures in methanol production, including case analysis of accidents in methanol production over the years, formulation of prevention and control measures, etc.

*3.3. Methanol Production Safety Information Ontology Knowledge Base*

Knowledge ontology is a system specification that analyzes objects in the domain and the relationship between objects, and explicitly, formally and in a way that can be shared, describes the concept of objects in the domain [26]. Knowledge ontology usually includes concepts (classes), properties, methods, instances, and other elements.

A formula was developed to consistently describe the ontology of the methanol production safety information. The formula sought to, analyze the methanol production safety information, obtain the specific classification of methanol production safety information knowledge ontology, and express the safety information knowledge ontology with knowledge quintuple:

$$S = \{P_1, \ P_2, \ H, \ A, \ R\} \tag{1}$$

According to formula (1), the production equipment knowledge ontology, process flow knowledge ontology, hazard source knowledge ontology, risk event knowledge ontology, and accident tree knowledge ontology are all part of the methanol production safety information knowledge ontology.

Where S (Safety information) represents the methanol safety production information;

$P_1$ (Production plant) refers to various production plant information involved in the methanol production process, including plant specification, quantity, operating conditions, etc.;

$P_2$ (Process) indicates the process information in methanol production;

$H$ (Hazard) represents the information on various hazard sources in the methanol production process, which is divided into Class I and Class II hazard sources;

$A$ (Accident tree) means to record the accidents that occurred in methanol production through the fault tree mode;

$R$ (Risk event) refers to the risk event existing in the methanol production process, including its risk factors, consequences, preventive measures, etc.

The five tuples of methanol production safety information knowledge represent the core concepts in methanol production knowledge. Through the further description of the category hierarchy relationships and properties relationships in the five tuples, the basic framework of the methanol production safety information knowledge base is constructed, and the ontology rule base is created by combining the characteristics and rules of safety information. The knowledge base stores the knowledge ontology categories and their properties and relationships of Production devices, Hazard sources, Accidents, etc., in the methanol production process, while the rule base stores the information reasoning rules. The knowledge base and rule base together constitute the methanol production safety information ontology knowledge base model, which completes the storage, sharing, and reuse of safety information.

## 4. Model Construction of Methanol Production Safety Information Knowledge Base

In this study, a methanol production safety information knowledge base model is established based on the conventional seven-step method and the characteristics of the safety information ontology knowledge base to be built. The process is divided into three parts: knowledge acquisition, knowledge representation, and knowledge storage and reuse, as shown in Figure 3.

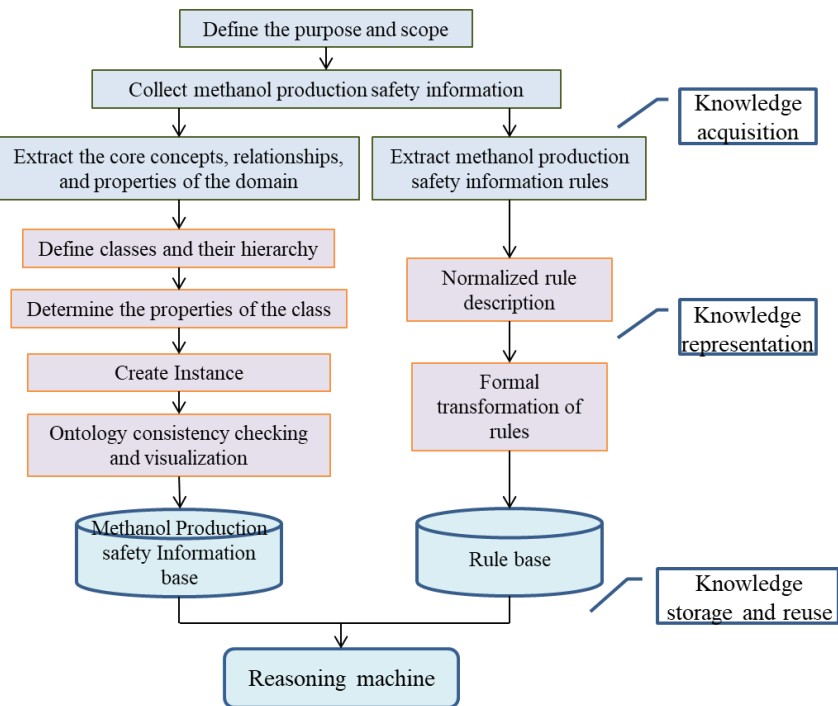

**Figure 3.** The construction process of the methanol production safety information ontology knowledge base model.

### 4.1. Knowledge Acquisition

(1) Specify the purpose and scope of the ontology created. The knowledge ontology created in this research is oriented to the field of safety information. Taking the methanol production process as the research object, it collects and sorts all kinds of safety information generated in the methanol production process, creates the methanol production safety information ontology knowledge base, stores and reuses safety information, and realizes the standardized description and expression of core concepts in the field.

(2) Collect methanol production safety information and knowledge. The methanol production process involves a wide range of safety information data. Strong safety information data must be backed up in order to create an exhaustive knowledge base. Through field investigation of the methanol production process site, the safety information related to methanol production such as device information, process flow information, production process parameters, and accident cases over the years is summarized and sorted.

(3) Extract the core concepts, properties, and relationships of the domain. The core concepts in the field of methanol production safety information are represented by the top level of the knowledge quintuple of $S = \{P_1, P_2, H, A, R\}$, and the next level concepts are extracted step by step in combination with the ontology characteristics of the knowledge elements. Property is the definition of the relationship between concepts and data characteristics. In this paper, a combination of manual and machine-based interpretation is used to extract knowledge.

(4) Extract methanol production safety information rules. In rule extraction, most people will choose the extraction method based on the rough set or neural network, but this method relies too much on algorithms and is not flexible enough, and the quality of extracted rules is not high. Therefore, this study adopts the method of manual rule extraction to extract the safety information rules of methanol production to ensure that the extracted rules are flexible and practical. We examine the extraction rules level by level from top to bottom in accordance with the hierarchy and property relationships of the developed knowledge ontology in order to assure the integrity of manual rule extraction. At the same time, to facilitate the subsequent reasoning, this study mainly extracts the

accident tree rules. For example, "If the top event T connected to the AND gate occurs, the basic events A and B connected to the AND gate will occur".

### 4.2. Knowledge Representation

#### 4.2.1. Define Class and Class Hierarchy

The safety information ontology class is determined according to the five tuples of safety information knowledge created, and the hierarchical structure of the class is determined in combination with the characteristics of methanol production safety information, as shown in Figure 4.

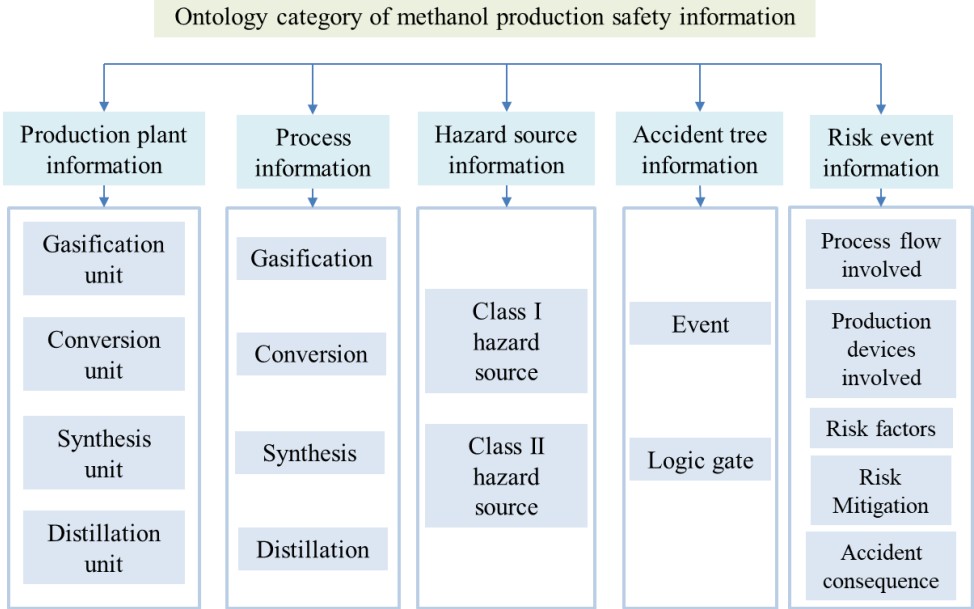

**Figure 4.** Classes of methanol production safety information ontology.

#### 4.2.2. Define Properties of the Class

Define the property of the safety information knowledge ontology class. Properties include object properties and data properties. The object property describes the association between the classes. The domain represents the starting point of the association, and the range represents the endpoint of the association. The class is connected with the class through the object property. The data property represents the data characteristic of the class and describes the data value type of the class. Tables 1 and 2 show some object properties and data properties of the methanol production safety information ontology.

**Table 1.** Some object properties.

| Serial No | Property | Domain | Range |
| --- | --- | --- | --- |
| 1 | Has Behavior | Operator | Unsafe Behavior |
| 2 | Cause Basic Event | Unsafe Behavior | Basic Event |
| 3 | Involve Risk Event | The Process flow | Risk Event |
| 4 | Has Measure | Risk Event | Preventive Measure |
| 5 | Logic Gate Species Is | Logic Gate | And Gate |
| 6 | Top Event Is | Logic Gate | Top Event |
| 7 | Basic Event Is | Logic Gate | Basic Event |
| 8 | Status Is | Top Event | Happen |

**Table 2.** Some data properties.

| Serial No | Property | Value Type |
| :---: | :---: | :---: |
| 1 | Spec (specification) | Varchar |
| 2 | Qty (quantity) | Int |
| 3 | Material | Varchar |
| 4 | The number of deaths | Int |
| 5 | The number of injured | Int |
| 6 | Property damage | Varchar |

4.2.3. Design Ontology Knowledge Rule Base

In the process of extracting safety information knowledge rules, the preliminary extracted rules often have problems such as unclear expression, logic confusion, and redundancy. In combination with the characteristics of the safety information itself, for example, when judging whether there is a hazard source, the basic rule of safety information can be described as "if there is···, then there is a ··· hazard source." Therefore, the production rule "IF···, THEN···" is used to express the safety information ontology rules in a standardized way. In the rule base, each rule may contain multiple premises and actions, that is, the form of ontology rules can be described as:

$$Rule = IF(Condition\ 1)AND \cdots (Condition\ n)$$

$$THEN(Action\ 1) \cdots (Action\ n)$$

For example, for the accident tree rule, it can be described as:

Rule = IF "The top event connected to the AND gate occurs", THEN "All the basic events connected to the AND gate will occur". The specific rule design will be detailed in the part of Knowledge Reuse and Reasoning Model Building.

For safety information ontology rules that cannot be described by production rules, process rules are used as supplementary expressions.

*4.3. Knowledge Storage and Ontology Visualization*

Considering ontology creation tools comprehensively, we choose the ontology editing and knowledge acquisition software based on the Java language, namely the Protégé software, as the ontology creation tool [27]. On the Java platform, Protégé is open-source software that may be used to build ontologies. To satisfy the varied demands of users, the software also offers a number of optional plug-ins, such as the SWRLTab plug-in, the OntoGraf plug-in, etc., which can implement reasoning rule design, ontology visualization, and other services. While using Protégé to build an ontology based on OWL (Web Ontology Language), you are not required to adhere to the OWL language writing specification, rather, you can simply concentrate on building ontology information. Strong compatibility, ease of use, and a clear user interface of Protégé software allow it to successfully meet a number of requirements while building a safety information ontology knowledge base. Protégé is utilized in this work to add class instances and build the methanol production safety information ontology. To complete the knowledge storage of methanol production safety information, the hierarchical structure of the methanol production safety information knowledge system is presented in the form of a tree directory.

The production equipment knowledge ontology, process flow knowledge ontology, hazard source knowledge ontology, accident tree knowledge ontology, and risk event knowledge ontology are the five components that make up the top-level class of the methanol production safety information ontology. Each top-level class also contains specific classes. For example, in the knowledge ontology of production equipment, the methanol production equipment is divided into gasification, conversion, sulfur recovery, synthesis, and rectification equipment, according to the different workshops. The next level includes equipment specifications, quantity, operating conditions, and other production status information. The class of hazard sources can be divided into Class I and Class II hazard

sources. The Class I hazard sources can be specifically divided into personnel, equipment, and systems. The Class II hazard sources can be divided into unsafe behavior of people, unsafe state of things, management defects, etc. Property addition includes object properties and data properties, defining the characteristics and associations of the added ontology.

After the ontology construction is completed, the pellet inference engine is used to complete the consistency check of the ontology to ensure the unity and correctness of the knowledge rules in the ontology. Figure 5 shows the visualization of the methanol production safety information ontology created by using the OntoGraf plug-in in Protégé.

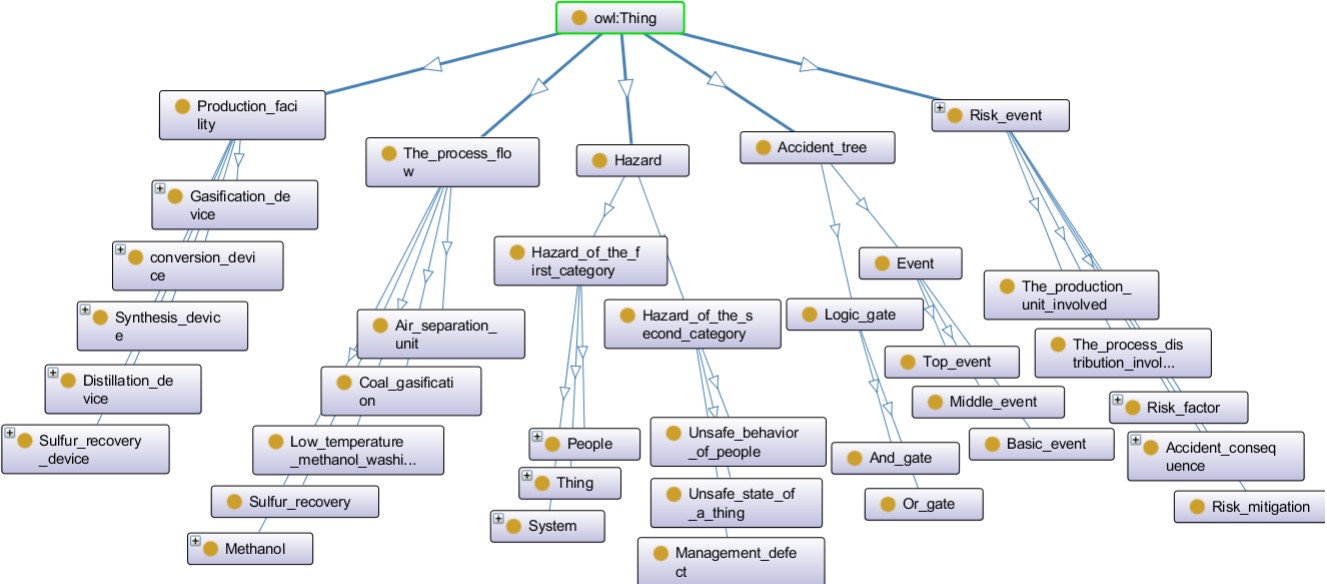

**Figure 5.** Visualization of methanol production safety information ontology.

## 5. Knowledge Reuse of Methanol Production Safety Information Ontology Knowledge Base

### 5.1. Reasoning Model Based on Accident Tree

Accident prevention is a key role of safety information. This part is based on the accident tree analysis theory, to reuse safety information to prevent accidents.

Figure 6 is an accident reasoning model based on the accident tree analysis theory. As shown in Figure 6, the basic process of accident tree reasoning is: by reading the created safety information ontology knowledge base, we can obtain the top event of the accident tree, judge the category of the logic gate connected to the top event, and then obtain the basic events connected to the logic gate, so as to obtain the basic events associated with the top event of the accident tree. It is convenient to check whether there are basic events occurring during the safety management process and to implement the necessary corrective actions to effectively avoid accidents from happening.

### 5.2. Design of Reasoning Rules for Accident Tree

The specific classes and object properties in the accident tree knowledge ontology must be expressed in depth in this section since we will be creating reasoning rules based on them. According to the notion of accident tree analysis, the accident tree is made up of events and logic gates. AND gates and OR gates are examples of logic gates. Top, middle, and basic events are all types of events. The accident tree knowledge ontology's specific classes are displayed in Table 3. The property relationships between different classes in the accident tree knowledge ontology are displayed in Table 4. Examples include the connection between logic gates and events, whether an event occurs or not, the type of event, the type of logic gate, etc.

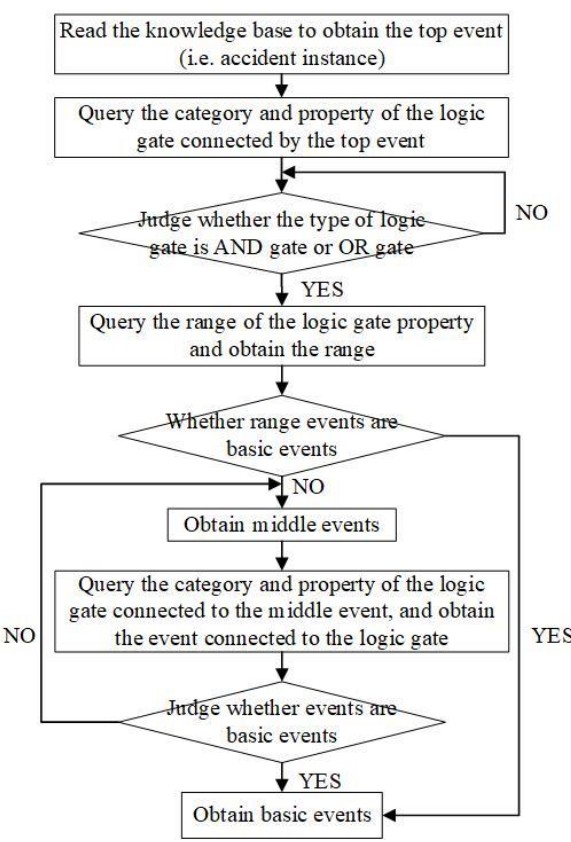

**Figure 6.** Accident tree reasoning flow chart.

**Table 3.** Accident tree ontology class information.

| Serial No | Class | Serial No | Class |
|---|---|---|---|
| 1 | Event | 7 | Logic Gate Species |
| 2 | Event Species | 8 | And Gate |
| 3 | Top Event | 9 | Or Gate |
| 4 | Middle Event | 10 | Event Status |
| 5 | Basic Event | 11 | Happen |
| 6 | Logic Gate | 12 | Not Happen |

**Table 4.** Accident tree ontology object property information.

| Serial No | Property | Domain | Range |
|---|---|---|---|
| 1 | Event Species Is | Event | Event Species |
| 2 | Logic Species Is | Logic Gate | Logic Gate Species |
| 3 | Status Is | Event | Event Status |
| 4 | Top Event Is | Logic Gate | Top Event |
| 5 | Middle Event Is | Logic Gate | Middle Event |
| 6 | Basic Event Is | Logic Gate | Basic Event |

*5.3. SWRL Rule Design and Transformation of Accident Tree*

SWRL rules can be used to design the reasoning relationships between each class instance in the ontology. The accident tree is composed of logic gates and events. By analyzing the process of accident occurrence, the basic events are gradually obtained from the top event. Therefore, we can contact SWRL reasoning rules to design accident reasoning rules, and obtain middle events, basic events, etc., from the top event, so as to obtain the causes of the accident.

(1) Based on the definition of the logical AND gate in the accident tree, the reasoning rule between the logical AND gate and events is designed and converted into SWRL rules, which are described as follows:

Rule1:

$$
\begin{aligned}
LogicGate(?x) \quad & \wedge TopEvent(?y) \wedge BasicEvent1(?z) \wedge \mathrm{BasicEvent}2(?a) \\
& \wedge LogicGateSpeciesIs(?x, AndGate) \wedge TopEventIs(?x, ?y) \\
& \wedge BasicEvent1Is(?x, ?z) \wedge BasicEvent2Is(?x, ?a) \\
& \wedge StatusIs(?y, Happen) \\
& \rightarrow StatusIs(?z, Happen) \wedge StatusIs(?a, Happen)
\end{aligned}
$$

In the above rules, $x$ is an instance of the class LogicGate, $y$ is an instance of the class TopEvent, $z$ is an instance of the class BasicEvent1, $a$ is an instance of the class BasicEvent2, and LogicGateSpeciesIs, TopEventIs, BasicEvent1Is, BasicEvent2Is, and StatusIs are object properties between defined classes.

The meaning of this rule is: if the instance $x$ is the AND gate in the logic gate, the top event of $x$ is $y$, the basic event 1 is $z$, the basic event 2 is $a$, and the top event y occurs, then we can conclude that event z and event $a$ occur.

(2) Based on the definition of the logical OR gate in the accident tree, the reasoning rule between the logical OR gate and events is designed and converted into SWRL rules, which are described as follows:

Rule2:

$$
\begin{aligned}
LogicGate(?x) \quad & \wedge TopEvent(?y) \wedge BasicEvent1(?z) \wedge \mathrm{BasicEvent}2(?a) \\
& \wedge LogicGateSpeciesIs(?x, OrGate) \wedge TopEventIs(?x, ?y) \\
& \wedge BasicEvent1Is(?x, ?z) \wedge BasicEvent2Is(?x, ?a) \\
& \wedge StatusIs(?y, Happen) \wedge StatusIs(?z, NotHappen) \\
& \rightarrow StatusIs(?a, Happen)
\end{aligned}
$$

The meaning of this rule is: if the instance $x$ is the OR gate in the logic gate, the top event of $x$ is $y$, the basic event 1 is $z$, the basic event 2 is $a$, and the top event $y$ occurs, then it can be concluded that event $a$ occurs.

Rule3:

$$
\begin{aligned}
LogicGate(?x) \quad & \wedge TopEvent(?y) \wedge BasicEvent1(?z) \wedge \mathrm{BasicEvent}2(?a) \\
& \wedge LogicGateSpeciesIs(?x, OrGate) \wedge TopEventIs(?x, ?y) \\
& \wedge BasicEvent1Is(?x, ?z) \wedge BasicEvent2Is(?x, ?a) \\
& \wedge StatusIs(?y, Happen) \wedge StatusIs(?a, NotHappen) \\
& \rightarrow StatusIs(?z, Happen)
\end{aligned}
$$

The meaning of this rule is: if the instance $x$ is the OR gate in the logic gate, the top event of $x$ is $y$, the basic event 1 is $z$, the basic event 2 is $a$, and the top event $y$ occurs, then it can be concluded that event $z$ occurs.

In OR gate reasoning rules, this paper defaults to the occurrence of the basic event with the highest probability of occurrence.

### 5.4. Example Demonstration—Fire and Explosion Accident Reasoning of Methanol Storage Tank (Area)

A fire and explosion accident of the methanol storage tank (area) is used as an illustration to confirm the validity and viability of the accident reasoning rules.

In the fire and explosion accident tree of the methanol storage tank (area), the top event is the fire and explosion event of the methanol storage tank, and the events leading to it include methanol leakage and an ignition source. Only when the event "methanol leakage" and the event "there is an ignition source" occur at the same time, the top event "methanol tank fire explosion accident" will occur, so the logic gate between the top event and middle events should be the AND gate. The logical connection between the middle

events and the basic events and the category of logic gates are also determined in this way, so we will not repeat them here. By analyzing the causes of the fire and explosion accident of the methanol storage tank step by step, the accident tree shown in Figure 7 is obtained.

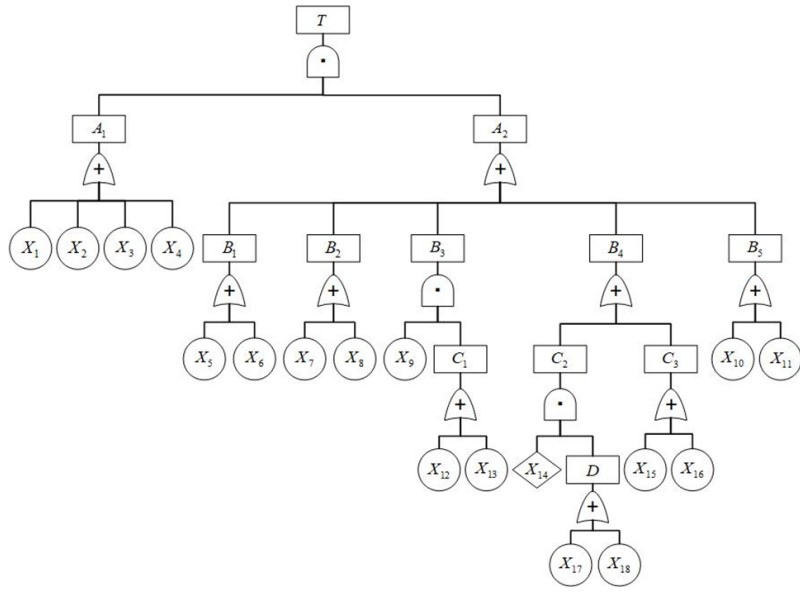

**Figure 7.** Methanol storage tank fire and explosion accident tree.

In order to make the representation of the methanol tank fire and explosion accident tree simple, we use letter codes to represent each event in the accident tree. Table 5 shows the specific event names represented by each letter code in the methanol tank fire and explosion accident tree in Figure 7.

**Table 5.** Accident tree event code.

| Event Code | Event Name | Event Code | Event Name |
|---|---|---|---|
| T | Fire and explosion of methanol storage tank | $X_4$ | Leakage caused by personnel operation error |
| $A_1$ | Methanol leakage forms mixed gas with air | $X_5$ | Smoking in the tank farm Species |
| $A_2$ | Presence of an ignition source | $X_6$ | Fireworks against rules in tank farm |
| $B_1$ | Open fire | $X_7$ | Use non-explosion-proof electrical equipment |
| $B_2$ | Electric spark | $X_8$ | Failure of explosion-proof electrical equipment |
| $B_3$ | Lightning spark | $X_9$ | Lightning strike |
| $B_4$ | Electrostatic spark | $X_{10}$ | Use iron tools |
| $B_5$ | Impact spark | $X_{11}$ | Wear shoes with nails |
| $C_1$ | Failure of lightning arrester | $X_{12}$ | Lightning protection grounding resistance exceeds the standard |
| $C_2$ | Static electricity exists in the tank | $X_{13}$ | Lightning protection grounding wire or grounding body is damaged |
| $C_3$ | Human's body carries static electricity | $X_{14}$ | Electrostatic accumulation in the tank |
| D | Unqualified static grounding device | $X_{15}$ | No anti-static work clothes |
| $X_1$ | Valve seal failure | $X_{16}$ | Contact with an ungrounded conductor during operation |
| $X_2$ | Flange seal failure | $X_{17}$ | Anti-static grounding resistance exceeds the standard |
| $X_3$ | Can body damaged | $X_{18}$ | The anti-static grounding wire or grounding body is damaged |

The above methanol tank fire and explosion accident tree instances were input into the created knowledge base and rule base model.

According to the above created accident tree reasoning rules, and combined with the methanol fire explosion accident tree, the accident reasoning rules can be designed, as shown in Table 6.

**Table 6.** Reasoning rules for fire and explosion accident tree of methanol storage tank.

| Serial No | Accident Rule | SWRL Rule |
|---|---|---|
| 1 | IF the event "methanol tank fire and explosion accident" occurs, THEN the event "methanol leakage forms mixed gas with air" and the event "ignition source exists" occur. | $LogicGate(?x) \wedge TopEvent\_T(?y)$ $\wedge MiddleEvent\_A1(?z) \wedge MiddleEvent\_A2(?a)$ $\wedge LogicGateSpeciesIs(?x, AndGate)$ $\wedge TopEventIs(?x, ?y) \wedge MiddleEvent1Is(?x, ?z)$ $\wedge MiddleEvent2Is(?x, ?a)$ $\wedge StatusIs(?y, Happen)$ $\rightarrow StatusIs(?z, Happen) \wedge StatusIs(?a, Happen)$ |
| 2 | IF the event "methanol leakage forms mixed gas with air" occurs, THEN at least one of the events "valve seal failure", "flange seal failure", "tank damage", and "leakage caused by human operation error" will occur (combined with specific examples, the event "valve seal failure" with the highest probability of occurrence will occur by default.). | $LogicGate(?x) \wedge MiddleEvent_A1(?y)$ $\wedge BasicEvent_X1(?z) \wedge BasicEvent_X2(?a)$ $\wedge BasicEvent_X3(?b) \wedge BasicEvent_X4(?c)$ $\wedge LogicGateSpeciesIs(?x, OrGate)$ $\wedge MiddleEventIs(?x, ?y)$ $\wedge BasicEvent1Is(?x, ?z) \wedge BasicEvent2Is(?x, ?a)$ $\wedge BasicEvent3Is(?x, ?b) \wedge BasicEvent4Is(?x, ?c)$ $\wedge StatusIs(?y, Happen)$ $\wedge StatusIs(?a, NotHappen)$ $\wedge StatusIs(?b, NotHappen)$ $\wedge StatusIs(?c, NotHappen)$ $\rightarrow StatusIs(?z, Happen)$ |
| 3 | IF the event "ignition source exists" occurs, THEN at least one of the events "open fire", "electrical spark", "lightning spark", "electrostatic spark", and "impact spark" will occur (combined with specific examples, the event "open fire" with the highest probability of occurrence will occur by default). | $LogicGate(?x) \wedge MiddleEvent_A2(?y)$ $\wedge MiddleEvent_B1(?z) \wedge MiddleEvent_B2(?a)$ $\wedge MiddleEvent_B3(?b) \wedge MiddleEvent_B4(?c)$ $\wedge MiddleEvent_B5(?d)$ $\wedge LogicGateSpeciesIs(?x, OrGate)$ $\wedge MiddleEventIs(?x, ?y)$ $\wedge MiddleEvent1Is(?x, ?z)$ $\wedge MiddleEvent2Is(?x, ?a)$ $\wedge MiddleEvent3Is(?x, ?b)$ $\wedge MiddleEvent4Is(?x, ?c)$ $\wedge MiddleEvent5Is(?x, ?d)$ $\wedge StatusIs(?y, Happen)$ $\wedge StatusIs(?a, NotHappen)$ $\wedge StatusIs(?b, NotHappen)$ $\wedge StatusIs(?c, NotHappen)$ $\wedge StatusIs(?d, NotHappen)$ $\rightarrow StatusIs(?z, Happen)$ |
| 4 | IF the event "open fire" occurs, THEN at least one of the events "smoking in the tank farm" and "illegal hot work in the tank farm" will occur (combined with specific examples, the event "illegal hot work in the tank farm" with the highest probability of occurrence will occur by default.). | $LogicGate(?x) \wedge MiddleEvent\_B1(?y)$ $\wedge BasicEvent\_X5(?z) \wedge BasicEvent\_X6(?a)$ $\wedge LogicGateSpeciesIs(?x, OrGate)$ $\wedge MiddleEventIs(?x, ?y) \wedge BasicEvent1Is(?x, ?z)$ $\wedge BasicEvent2Is(?x, ?a)$ $\wedge StatusIs(?y, Happen)$ $\wedge StatusIs(?z, NotHappen)$ $\rightarrow StatusIs(?a, Happen)$ |

As seen in Figure 8, the designed accident tree reasoning rules were entered into the SWRLTab plug-in in Protégé.

The "Run Drools" button was pressed to start the rule engine after entering the defined reasoning rules. The reasoning rules were run. The deduced axioms were converted into OWL knowledge by pressing the "Drools ->OWL" button. The conclusion of the reasoning can be seen in Figure 9.

According to the reasoning results displayed in Figure 9, the middle causes of the methanol tank fire and explosion accident are "mixed gas formed by methanol leakage and air" and "presence of ignition source," and the basic causes are "valve sealing failure" and "illegal hot work in the tank farm." Therefore, targeted measures can be taken to prevent

fire and explosion accidents of methanol tanks, such as inspection and maintenance of valves of ineffective tanks and no illegal hot work in the tank farm.

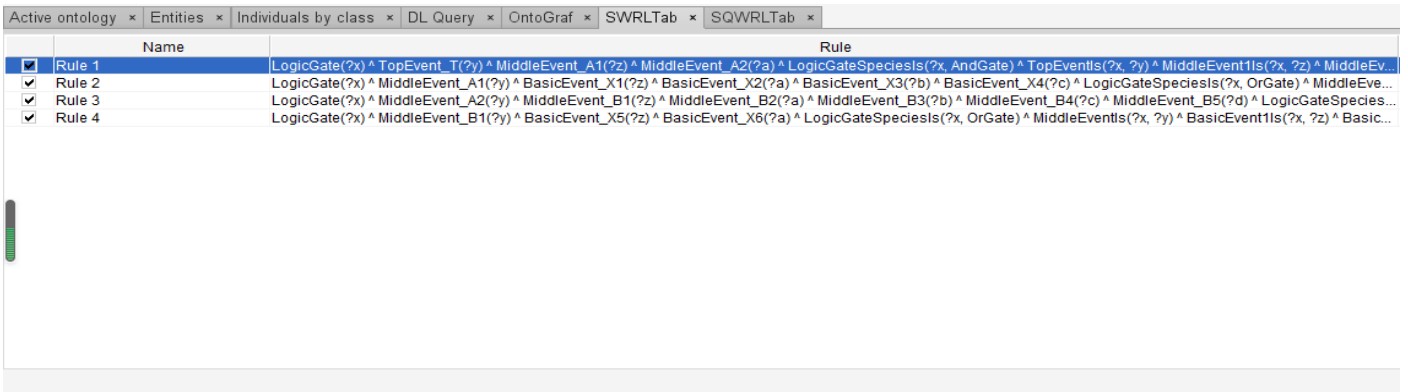

**Figure 8.** Accident tree SWRL rule entry diagram.

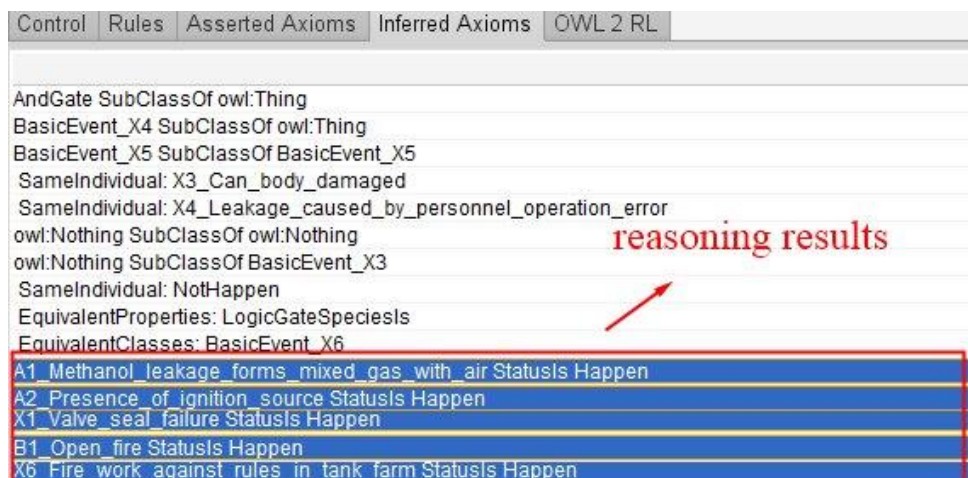

**Figure 9.** Accident tree reasoning results.

The case reasoning of the methanol tank fire explosion accident tree shows that the reasoning rules can deduce the important accident causes from past occurrences, allowing for the implementation of efficient preventative measures. The reasoning rules' precision and applicability are confirmed.

The results of the aforementioned reasoning are either fresh knowledge mined by reasoning that is not present in the knowledge base or the knowledge concealed in the ontology. This demonstrates the enormous benefit of combining ontology and rules in the field of safety information. On the basis of the accident tree ontology in the created safety information knowledge base, the association between various classes in the accident tree knowledge ontology is established by designing the accident reasoning rules, and the middle events and basic events related to the top event are obtained by reasoning, that is, mining the hidden knowledge in the accident information and identifying the cause of the accident. The knowledge base model for methanol production safety information ontology that was created can not only store and manage the safety information but also obtain the deep knowledge implied in the safety information and dig out new knowledge that does not yet exist.

## 6. Conclusions

An accident tree reasoning model and an ontology-based safety information knowledge base model of hazardous chemical companies are built in this work to help safety managers with safety information management and accident prevention utilizing safety information. The main results and contributions of this study are as follows:

Firstly, the establishment of a methanol production safety information knowledge base realizes the unified representation and classified storage of methanol production safety information. Taking methanol production as the research object, the safety information of hazardous chemical enterprises is deconstructed and analyzed, and the knowledge quintuple of $S = \{P_1, P_2, H, A, R\}$ is constructed. The safety information is divided into production equipment knowledge ontology, process flow knowledge ontology, hazard source knowledge ontology, accident tree knowledge ontology, and risk event knowledge ontology. The hierarchical structure and property relationships between ontology classes are defined, and then the ontology-based methanol production safety information knowledge database is constructed. It can efficiently process huge amounts of safety information data by storing and managing complicated safety information in a consistent manner, as well as realizing ontology visualization with the Protégé software, which addresses the issue of a lack of integrity and systematization in the administration of safety information. In the area of hazardous chemicals, the safety information knowledge base model can provide theory and practice references for the storage, representation, and management of enterprise safety information data.

Secondly, based on the accident tree analysis theory, the reasoning rules for accidents are developed using the created accident tree ontology. The reasoning rules are then translated into the semantic web rule language, and the causes of the accident are inferred by the inference engine. The case reasoning of the methanol tank fire and explosion accident demonstrates the accuracy and applicability of the established accident reasoning rules. The reuse function of safety information is realized by the accident tree reasoning model, which is based on the safety information ontology knowledge base and rule base. As a result, accident tree ontology reasoning can be successfully used for safety information reasoning. On the one hand, the causes of accidents can be uncovered to effectively prevent production accidents, on the other hand, it can significantly enhance the semantic reasoning capability of the safety information knowledge base, uncover previously undiscovered knowledge and new knowledge, increase the sharing rate and reuse rate of safety information, and has a high flexibility and scalability.

However, the research on the combination of ontological reasoning rules and accident tree analysis theory in this paper is still in the exploratory stage. How to build a more applicable and comprehensive accident tree ontology and accident reasoning rules still needs more in-depth research. In our future work, we will focus on how to mine all OR gate events according to the probability of occurrence of events; how to use ontology and SWRL rules to quantitatively analyze the accident tree; how to construct ontology and reasoning rules for dynamic accident trees; and how to design the reasoning rules between the accident tree ontology and other ontologies, such as hazard source ontology and risk event ontology, etc.

**Author Contributions:** Conceptualization, M.L. and R.H.; methodology, M.L.; software, M.L.; validation, M.L. and F.X.; formal analysis, M.L. and R.H.; investigation, M.L.; resources, M.L. and Rui Huang; data curation, M.L.; writing—original draft preparation, M.L.; writing—review and editing, M.L., R.H. and F.X.; visualization, M.L.; supervision, F.X.; project administration, M.L. and F.X.; funding acquisition, M.L. and R.H. All authors have read and agreed to the published version of the manuscript.

**Funding:** This work was funded by the National Key R&D Program of China (2018YFC0808406) and the Fundamental Research Funds for the Central Universities of Central South University, grant number 2022ZZTS0487.

**Institutional Review Board Statement:** Not applicable.

**Informed Consent Statement:** Not applicable.

**Data Availability Statement:** Not applicable.

**Acknowledgments:** The authors are grateful to all study participants.

**Conflicts of Interest:** The authors declare no conflict of interest.

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
