# Peer review of "Research on the Construction of Safety Information Ontology Knowledge Base and Accident Reasoning for Complex Hazardous Production Systems-Taking Methanol Production Process as an Example"

_sustainability, doi:10.3390/su15032568_

Round 1

Reviewer 1 Report

The article “Research on the Construction of Safety Information Ontology Knowledge Base and Accident Reasoning for Complex Hazardous Production Systems ” takes methanol production as an example, the concept of “ontology” is introduced to construct the safety knowledge ontology, and the safety information knowledge base is created with the help of protégé software. And the cause of the accident using accident tree reasoning and mine the vast knowledge of safety information. The article has some shortcomings in handling some details. Here are my opinions on the article:

 (1) Please indicate the full name of the abbreviations first appearing in the article, such as IBM in line 26.

(2)Please explain why the methanol production process is selected as the research object.

(3)Please specify the classification basis of methanol safety information in Line 81.

(4)Please mark 98 lines of formula and explain the basis for establishing the formula.

(5)149-151 lines, how to ensure the integrity of information extraction depends on the method of manual rule extraction. Please explain it .

(6)Please explain the reason for choosing protégé software.

(7)Please explain how Tables 3 and 4 are derived.

(8)Please explain what the SWRL rule means in line 271, why it be combined with FTA.

(9)Figure 7 and Table 5 do not appear in the text, please check.

(10)368-371 lines, The knowledge base... does not yet exist., please explain how to dig out the deep knowledge implied in the safety information.

Reviewer 2 Report

The authors have done a lot of work on the construction of safety information ontology knowledge base. Somehow, the manuscript in its present form is not ready for publication. I could recommend the three changes to make your work more attractive to readers. First, it is suggested to specify the dangerous chemical companies in the title. In the study, a methanol production process is taken as an example to build the ontology-based safety information knowledge base model of hazardous chemical companies, which cannot represent the whole complex hazardous production system. Second, please provide sufficient background and more relevant literature review. Third, it is recommended to improve the appearance of the figure, such as the font in Figure 3, and the format in Figure 4, Figure 5 and Table 5.

Reviewer 3 Report

The article entitled " Research on the Construction of Safety Information Ontology Knowledge Base and Accident Reasoning for Complex Hazardous Production Systems” is well organized and provides a lot of data.

However, there is a shortcoming that must be addressed before publication:

I think it would be good to include the information about Risk Mitigation in Figure 4 (line 218).

I also consider it appropriate to discuss the information about Risk Mitigation within the work.

Reviewer 4 Report

The article deals with an interesting issue which is the problem of the occurrence of hazards in technological processes, in this case the production of methanol.

1. Introduction

- lines 34-36 - The Authors wrote that enterprise management is hampered by the loss of a significant amount of useful safety information due to imperfect information processing methods - What safety information may be lost? According to the Authors, what methods of information processing are imperfect?

- line 49 - error in writing, should be Y.T.

- lines 53-54 - The Authors wrote that the analysis of the cited literature shows that the ontological knowledge base fulfills information management functions such as unity, professionalism and cooperation - it is hard to say due to the fact that the authors gave only examples of application. It is also worth referring to specific sources of literature, and not pointing to the cited literature.

The review of the literature does not refer to the issue of accidents, the methods used to assess the risk of a dangerous event or process risk assessment.

2. Review of the knowledge base on the safety of methanol production

- it would be worth introducing the Material and Method chapter in the article and dividing the information from chapters 2-4 compiled in the study.

 - line 75 - worth adding (Figure 1).

 - line 115 – no capital letter.

3. Model Construction of Methanol Production Safety Information Knowledge Base

- line 155 – no capital letter.

- lines 169-170 - it is worth referring to table 1 and table 2 in the text.

4. Knowledge (…)

- lines 261-264 - The Authors point to corrective actions or only such corrective actions will effectively prevent accidents?

The Authors write about the SWRL rule and the transformation of the accident tree without justifying its use.

5. Conclusions

Conclusions refer to the summary of the study, no discussion of the results, no directions for further research, no purposefulness of using the methods that are the subject of the work.

Round 2

Reviewer 1 Report

It has been revised according to the review comments.

Reviewer 4 Report

Thanks to the Authors for the detailed answers. In the study, I would also pay attention to Fig. 5 (there are two drawings, it would be worth describing each).